# Influence of Boiling Time on Chemical Composition and Properties of Tender and Mature Moringa Pods

**DOI:** 10.3390/foods13121823

**Published:** 2024-06-10

**Authors:** María Luisa Castelló, Tomás Sesé, Francisco José García-Mares, María del Sol Juan-Borrás, María Dolores Ortolá

**Affiliations:** 1Food Engineering Research Institute—FoodUPV, Universitat Politècnica de València, Camino de Vera s/n, 46022 Valencia, Spain; tosegui@etsiamn.upv.es (T.S.); majuabor@upv.edu.es (M.d.S.J.-B.); mdortola@tal.upv.es (M.D.O.); 2Department of Hydraulic Engineering and Environment, Universitat Politècnica de València, Camino de Vera s/n, 46022 Valencia, Spain; fjgarcia@gmf.upv.es

**Keywords:** moringa pods, boiling, protein, color, phenolic profile, antioxidant capacity

## Abstract

*Moringa oleifera* is a plant native to India that is well adapted to warm climates with a high yield and low agronomic requirements. Pods are one of the edible parts of this plant and are commonly consumed in some places, (India, Morocco, etc.) when in an early vegetative state. However, both production and consumption of this plant are scarce and seasonal in Europe and treatments to extend its shelf life are required. Therefore, the aim of this study has been to evaluate the variation in the physicochemical properties of *Moringa oleifera* pods at two stages of maturity, tender and mature, in terms of mass variation, optical and mechanical properties, protein content, total antioxidant capacity and phenolic profile, after boiling them in tap water at 100 °C for different lengths of time (0, 2, 5, 8, 12, 16, and 20 min). The mass of the tender pods increased by 15% during cooking, while the mature pods gradually lost weight. The protein content was approximately 4% with no significant change brought about by cooking. Ferulic, *trans*-cinnamic, *p*-coumaric, and ellagic acids were found in the fresh pods. During cooking, these phenols disappeared, and others appeared, including epicatechin and quercetin 3-glucoside, especially in the tender pods. In conclusion, boiling could contribute to an improvement in the organoleptic properties of moringa pods and to an extension of their storage and to wider availability on the market.

## 1. Introduction

*Moringa oleifera*, usually known as the drumstick or horseradish tree, is a fast-growing plant cultivated in warm climates with low rainfall and characterized by low agronomic requirements. These conditions are native to the southern Himalayas, north-eastern India, Afghanistan, Pakistan, and Bangladesh, where moringa originates. Each part of the moringa can be used, from the leaves and flowers, to the fruits and roots, and can be put to different purposes. They can be used as food for humans and animals, as a medicine, as an ornament, as protection for other crops, as a means of purifying water and obtaining oils, as fertilizers, or for the purposes of biomass production.

The moringa fruit is in the form of pods which are capsular, linear, and pendulous, measuring up to 40 cm in length and about 2 cm in width. Young, green moringa pods are usually harvested about 40 days after flowering, as a woody husk forms during the ripening process. The unharvested pods continue to ripen and thicken on the tree for a further 3–4 months until they reach senescence and give way to a new flowering. They usually contain 10–12 seeds per fruit. In India, pod production has been estimated at 19 kg/tree/year, and selected varieties in Hawaii yield 3–8 times more [1]. The pods are a darker, more intense green when they are at a less developed stage, and a lighter green when they are more mature. Once the fruit is in its mature stage, it shows irregularities as a consequence of the thickening of the seeds. Moringa pods contain numerous nutrients, such as proteins, containing the most essential amino acids, vitamins, and minerals [2]. They are an excellent source of vitamins, particularly vitamin C, vitamin A, B-complex vitamins, and vitamin K. These vitamins play a crucial role in various physiological functions of the human body, including immunity, vision, bone health, and metabolism [3]. Moringa pods provide a rich supply of critical minerals like calcium, iron, potassium, and magnesium. These minerals are essential for maintaining healthy bones, supporting muscle function, facilitating the formation of red blood cells, and regulating blood pressure, among other vital bodily processes [3]. Immature pods contain around 47 g crude fiber/100 g [4,5] and 21 g crude protein/100 g on a dry weight basis [4,5,6]. Matured pod contain 46 ± 1 g total dietary fiber/100 g and 17 g protein/100 g [7,8]. 

Moringa pods can be used to feed both humans and animals or for medicinal purposes. Their high amino acid and protein content and low level of anti-nutritional compounds make the pods highly valued for human consumption. Their taste is similar to that of asparagus, and they are consumed (mainly in India) in stews, boiled or pickled in a similar way to green beans or peas [9,10,11,12]. However, moringa pods are cooked at home, not processed commercially [13]. The main modifications of the product depending on the cooking conditions are unknown. In addition, the flour obtained after drying the pods is also used to improve baked products, fried products [2], pasta [14], bread [15], and instant soup [16]. Although the pods’ protein levels are lower than those of the seeds and leaves, their high fiber content and their fats, minerals, vitamins, and other bioactive compounds make them an optimal raw material for supplementing animal feed [7,17]. Due to the high fiber content of the pods, their consumption is indicated for treating digestive problems and preventing colon cancer [4]. Studies conducted by [18] showed that boiled pods had chemopreventive properties, reducing the appearance and multiplication of colon tumors in mice due to the presence of bioactive components, such as niazimycin and glucomorimgine. Besides, several studies have reported that moringa pods positively affect the anti-inflammatory activity of the gut [19,20]. On the other hand, scientific studies have shown that certain cooking treatments soften cell walls, thus facilitating the extraction of certain bioactive compounds such as polyphenols [21]. Furthermore, it appears that the type of cooking, such as frying or boiling, has a different influence on these bioactive compounds. In this way, [18] proved that the main bioactive compounds and antioxidant activities of the leafy vegetables analyzed were reduced when they were fried. Conversely, if they were steamed or boiled, they showed higher levels of polyphenols, flavonoids, and antioxidant capacity than fresh leaves. Therefore, it is necessary to understand the impact that cooking vegetables has on their antioxidant activity or free radical capacity. Cooking techniques, together with the proportion of ingredients, may modify several properties of the product, such as its composition, optical and mechanical properties, and flavor. Thus, it is vital for both scientific research and the consumer to know how to prepare and cook vegetables in order to obtain the best textural attributes during mastication [22].

Despite the nutritional value of *Moringa oleifera* pods, there have not been many previous studies that have evaluated the changes in their properties brought about by cooking. Thus, the main objective of this study is to evaluate the variation in the physicochemical properties of *Moringa oleifera* pods at differing stages of maturation (tender and mature), at different boiling times. This could offer new consumption options for vegetable products with high nutritional value, whose production is sustainable and adapted to climate change.

## 2. Materials and Methods

### 2.1. Raw Material

The experiments were carried out on moringa pods (*Moringa oleifera*). These fruits were harvested from the fields of the Universitat Politècnica de València, (Valencia, Spain) between the months of November and April, during the 2021–2022 campaign. Two stages of ripening were obtained: tender and mature (Figure 1).

### 2.2. Boiling the Moringa Pods

A cooking process was chosen which resembled the boiling of vegetables such as green beans or asparagus, at 100 °C for around 20 min. A previous study was performed to check that they were edible after that time. Previous times were also applied to obtain kinetic information regarding the changes which took place in the boiling process of moringa pods. Raw moringa pods were considered as the control samples. Thus, the pods were cut into strips 5 cm in length and were boiled in tap water at 100 °C for 0, 2, 5, 8, 12, 16, and 20 min; this was performed three times for each pod. The diameter was about 2 cm for mature pods and 0.5 cm for tender pods. Figure 2 shows how the boiling treatment was carried out along with some pictures showing the shapes of the tender and mature pods. 

### 2.3. Analytical Determinations

The water content of raw and boiled pods was determined by a gravimetric method [23]. The samples were pre-dried in an oven at 60 °C for 24 h and then dried in a vacuum oven (J.P SELECTA, Conterm model, Barcelona, Spain) at 60 °C for 48 h until a constant weight was reached. It was expressed in water mass fraction, and the soluble mass fraction was obtained by its difference to 1. 

Water activity (a_w_) was measured with a dew hygrometer (METER Aqualab 4TE, Benchtop Water Activity Meter, Aqualab, Pullman, WA, USA) following method 978.18 [24].

The protein content was analyzed using the Kjeldhal method [25] based on the amount of nitrogen, by digestion, distillation, and titration (Block digest 6, J.P. Selecta, Barcelona, Spain; Distillation unit UDK 127, Velp Cientifica, Usmate Velate, Italy). The protein percentage of the samples was calculated using 6.25, the standard value for vegetable products, as the conversion factor.

The CIEL*a*b* coordinates of pods were determined using a spectrophotometer (“Konica Minolta” Inc. Model CM—3600d, Tokyo, Japan) with a reference illuminant D65 and a 10° observer [26]. Moreover, differences in the color of boiled pods with regards to raw material in both cases, the mature and tender stages, were also recorded applying the following equation:$$\Delta E=\sqrt{{\Delta L}^{2}+{\Delta a}^{2}+{\Delta b}^{2}}$$

In order to evaluate the mechanical properties of moringa pods, a puncture test was carried out with a 2 mm diameter puncture probe using a universal press (Texturometer TA-XTplus Texture Analyse Aname, Stable Micro Systems, Godalming, UK), by analyzing the maximum force and the area of the test curve [27]. 

Furthermore, the total antioxidant capacity was determined by the DPPH (2,2-diphenyl-1-pyrcyrylhydrazyl) method, measuring the difference in absorbance using a spectrophotometer at a wavelength of 515 nm [28] and expressing the results as Trolox equivalents per g of moringa pods. This analysis was carried out on fresh mature and tender pods, as well as when they had been cooked for 5 and 20 min, as representative times of the heat treatment applied. Additionally, the phenols present in the samples were analyzed using an HPLC system, Agilent 1200 Series Rapid Resolution, equipped with a diode array detector, Agilent 1260 Infinity II (Agilent, Palo Alto, CA, USA), following the methodology described by [29]. Chromatographic separations were carried out on a Kinetex column (250 × 4.6 mm I.D., 5 μm; Phenomenex; Torrance, CA, USA) maintained at 31 °C. The binary mobile phase consisted of phase A (1% formic acid) and phase B (acetonitrile) with a gradient elution program (Table 1) applied at a constant flow rate of 0.5 mL/minutes. The sample inject volume was 10 mL. The spectral data from every peak were stored in the range of 190−500 nm, and the phenols were monitored at specific wavelengths: 230, 250, 260, 280, 290, 320, 350, and 380 nm. Analytical data acquisition and processing were determined using Agilent MassHunter Workstation software (version B.09.00).

Each compound was identified by comparing chromatographic retention times and UV spectral characteristics with those of representative standards: gallic acid, 4-*O*-caffeoylquinic acid, caffeic acid, rutin, ellagic acid, *p*-coumaric acid, ferulic acid, quercitrin, apigenin 7-glucoside, quercetin, *trans*-cinnamic acid, naringenin, vanillic acid, 4-hydroxybenzoic acid, epicatechin, quercetin 3-glucoside, sinapic acid, and kaempferol. The calibration curve (0.5, 1, 3, 6, 10, 20, and 40 mg/L) of each compound was plotted through the linear regression analysis of the area in each chromatogram versus its nominal concentration. The quantification of each compound, in order to reach the maximum sensitivity, was carried out in the chromatogram and recorded at the wavelength in which it presented its maximum absorption: 250 nm for vanillic acid; 260 nm for 4-hydroxibezoic acid, quercetin 3-glucoside, and quercitrin; 280 nm for gallic acid, epicatechin, and *trans*-cinnamic acid and 320 nm for 4-*O*-caffeoylquinic, caffeic acid, *p*-coumaric acid, ferulic acid, and apigenin-7-glucoside.

All of these measurements were conducted at least thrice.

### 2.4. Statistical Analysis

Statgraphics Centurion XIX.64 software was used for the statistical analysis of the results. An ANOVA analysis of varia nce was performed, using the LSD (Least Significant Difference) test at a significance level of 95% (*p*-value ≤ 0.05).

## 3. Results and Discussion

Figure 3 shows the results of water and solute mass fractions along with water activity, as well as the total mass variation in the tender and mature moringa pods subjected to the cooking process for different lengths of time. As can be seen, the water and solute contents in both types of pods were very similar throughout the cooking process. However, the tender pods recorded an increase of around 15% in mass during cooking, while the mature pods gradually lost weight. This behavior would be related to how the structure of the tissues evolves in line with the maturation of the fruit, which would have less water retention capacity as ripening progresses. In addition, the difference in the dimensions of the pods depends on the maturity stage and could affect the cooking kinetics. As was expected in a product with such a high amount of water, the water activity was also close to 1, thus meaning that its shelf life will be shorter. Therefore, some preservation processes (canning with subsequent sterilization or freezing) should be applied to commercialize it.

The protein content of fresh and boiled pods (Figure 4) was similar, with no significant differences due to the effect of cooking time or the maturity stage of the pods. The average value was 4.0 ± 0.7 g of protein/100 g pod, which is slightly lower than that reported by other authors, i.e., 7.1 g/100 g [30] and 10.1 g/100 g [7]. Compared with other vegetables consumed in a similar way (cooked), moringa pods are richer in protein. For example, green asparagus contains ≈2.9% [31] and green beans 1.97% [32].

Figure 5 shows the representation in the chromatic diagram of the a* and b* coordinates and their variation throughout the boiling and the values of luminosity (L*), chrome (C*), and hue (h*). Furthermore, the difference in color (ΔE) with regards to raw material in both cases, mature and tender pods, is represented.

In the case of the tender pods, the boiling time did not alter the L* of the sample at any time, which was also observed by [33] during the blanching of fresh broccoli, carrots, and green beans. In the case of the mature pods, boiling increased this parameter with respect to the raw pod, but it remained constant over time. a* and b* coordinates are in the second quadrant (green-yellow zone). Throughout the cooking, the pods were observed to behave differently depending on the maturity stage. Thus, when the mature pods were cooked, they lost color purity, whereas, when tender, their color chrome increased during cooking. This has been related to a modification in the surface-reflecting properties caused by the replacement of intercellular air with cell juice [34]. On the other hand, the hue of the cooked pods increased during the first 5 min of boiling, and then decreased until it remained stable between 12 and 20 min of heat treatment regardless the stage of ripening of the samples. Furthermore, changes in composition during heating process can cause oxidation of components or pigments which would contribute to the color evolution. Differences of color were significantly higher in mature boiled pods (ΔE ≈ 15) than in tender pods (ΔE ≈ 5), which indicates the higher appearance stability of these last samples. Moreover, initially, the differences of color of raw material was very close to 2, thus meaning that the color is not distinguishable by the human eye.

Figure 6 shows the values obtained in the puncture test of the two types of pods, both fresh and after different cooking times. In the case of the results obtained for maximum strength, it was observed that the hardness remained constant throughout the cooking times in the tender pods, while in the mature pods, there was a decrease in strength during the first 5 min of cooking, indicating a decrease in firmness with a degree of softening of 33%. This behavior is consequence of the initial differences in terms of hardness between both the raw samples, resulting in the tender pods being much softer than the mature pods. On the other hand, as regards to the area, a progressive decrease was observed in both types of pods throughout cooking with respect to the raw pods. The tender pods being those with the lowest values of this parameter, evidencing a loss of consistency. These results are coherent with those obtained by [22] who cooked for 10 min, maintaining a pressure of 2 psi with different ratio pods:water and salt concentration. The values of the hardness of their raw material (19.11 N) were very similar to our mature pods. They reported that the pressure applied during cooking on moringa pods caused the loss of the cell walls leading to accumulation of more water in the cells, which softens the texture of the product.

Figure 7 shows the antioxidant capacity of tender and mature pods, expressed in milliequivalents of Trolox per gram of pod, using the DPPH method described in the materials and methods section. The greater antioxidant activity of the tender pods (109 ± 34 mg Trolox/g pod), before cooking, compared to the mature pods (23 ± 19 mg Trolox/g pod) is noteworthy, with few significant differences as a result of the cooking time. Only in the mature pods did the antioxidant capacity increase after long cooking times, equaling their values with the values of tender pods. Therefore, maturity stage of pods would not influence the contribution of total antioxidant to the consumer of this boiled product. These results are contrary to those reported by [35], who observed a decrease of approximately 10% in antioxidant activity in moringa pods boiled at 100 °C for 30 min. This may be due to the longer time of the treatment in comparison with the conditions applied in this research. Even though they concluded that since a trivial amount of nutrient content was diminished by this heating treatment, providing cooked pods could provide significant nutritional benefits to individuals. However, studies by [33] demonstrated that the blanching of broccoli, carrots, and green beans did not lead to there being any significant differences with regards to their antioxidant capacity. The retention of antioxidant activity brought about by blanching could be related to the development of enzymatic reactions (PPO) [33]. Furthermore, according to [34] an overall increase of total antioxidant capacities was observed in cooked carrots, courgettes, and broccoli by different methods (boiling, steaming, and frying). This may be because of matrix softening and increased extractability of compounds, which could be partially converted into more antioxidant chemical species. Their findings defy the notion that processed vegetables offer lower nutritional value.

Table 2 shows the amount (average and standard deviation, expressed as mg/100 g) of the phenolic components of moringa pods considering their stage of ripening (mature or tender) and the effect of boiling time. As can be seen, these components were classified into phenolic acids and flavonoids. Of the 18 compounds considered in the samples analyzed, only 13 were detected and quantified. As an example, an HPLC chromatogram is shown in Figure 8 with the phenolic profile of the moringa pod. In order to facilitate the visualization of the changes that have occurred in the tender and mature moringa pods and the effect after cooking, Figure 9 shows in a bar chart the evolution of the 9 phenolic acids and 4 flavonoids quantified in the moringa pods samples. The results indicate that the tender pods had a higher concentration of phenols than the mature, as can be seen from the greater antioxidant activity shown above. The antioxidant capacity of vegetables has been mainly attributed to their phenolic compound content [15]. This fact has been reported by other studies; [36] assessed the effect that cooking has on the antioxidant properties in different varieties of green beans. They concluded that the factors involved in the changes in the polyphenol content were the same ones that caused changes in their antioxidant activity.

It is generally observed that although phenolic acids were more abundant in number and although they were not highly concentrated since the flavonoids epicatechin and quercetin 3-glucoside were the ones with the highest concentrations, this fact was more marked in the tender pods. This fact is more noticeable when the pods were boiled for 5 to 10 min.

Concretely, in fresh pods, only 4 phenolic acids were detected: ellagic, *p*-coumaric, ferulic, and *trans*-cinnamic acid. Although more compounds were detected (gallic, chlorogenic, caffeic acid, kaempferol, quercetin, rutin, catechin, and syringic acids) in studies published by other authors [8], 3 of the 4 phenols detected in this analysis do coincide (ellagic, *p*-coumaric, and ferulic acids). However, the concentrations are not comparable, probably due to the extraction and analysis process itself. Cooking the tender and mature moringa pods for up to 20 min caused the disappearance of the phenols present in the raw pods and the appearance of others, notably epicatechin, quercetin 3-glucoside, quercitrin, and apigenin 7-glucoside. It should be noted that *trans*-cinnamic phenolic acid is the only one of the 4 acids present in all the raw pods that does not disappear after cooking for 5 and 10 min, even increasing its concentration in both cases. Specifically, the concentrations of epicatechin and quercetin 3-glucoside in pods cooked for 5 or 10 min were very high: 52.33 ± 0.08 mg/100 g and 10.38 ± 0.05 mg/100 g, respectively. The flavonoid epicatechin is the one with the highest concentration by far, especially in the tender pods. In general, cooking for 20 min resulted in a lower concentration of the phenols recorded in both pods. This could be related to the fact that polyphenols are very unstable and highly susceptible to degradation with some processes which involve high temperatures [37]. These results are barely comparable with other published studies, since no research has been found into the evolution of phenols in cooked moringa pods. In other fresh vegetable pods, such as green or broad beans, catechin (0.41 and 16.23 mg/100 g, respectively), and epicatechin (0.69 and 37.55 mg/100 g, respectively) [38] had values that were much lower than those recorded in the cooked moringa pods used in this study. Cooking the moringa pods clearly modified the phenolic profile varying the contents of native polyphenols.

## 4. Conclusions

Boiling could improve the organoleptic properties of moringa pods and increase their storage and market availability. Consequently, offering new consumption formats for moringa could boost the production of this crop that is more adapted to climate change than others. In terms of mass variation during this treatment, the opposite behavior of the pods as a function of their maturity stage was relevant. Thus, the tender pods gained weight while the mature pods lost weight proportionally to the treatment time. This could be a consequence of the structural evolution of the moringa pod tissue with maturation, which would result in a lower water retention capacity. Although protein content did not vary with pod maturity or cooking time, the optical and mechanical properties as well as the phenol content were affected by these parameters. In particular, mature pods showed the highest differences of color regardless of the boiling time. They also lost color purity when they were cooked, unlike what happened to tender pods. Furthermore, only mature pods registered a decrease in firmness with a degree of softening of 33% in the first 5 min of the treatment. According to this study, it can be said that moringa pods have a greater quantity of phenolic acids than flavonoids, although the latter are present in much higher concentrations. To be specific, ferulic, *trans*-cinnamic, *p*-coumaric and ellagic acids were found in the fresh pods. During cooking, these phenols disappeared and others appeared, including epicatechin and quercetin 3-glucoside, especially in the tender pods. Finally, further studies should be performed to evaluate the sensorial acceptance of these pods and their shelf life.

## Figures and Tables

**Figure 1 foods-13-01823-f001:**
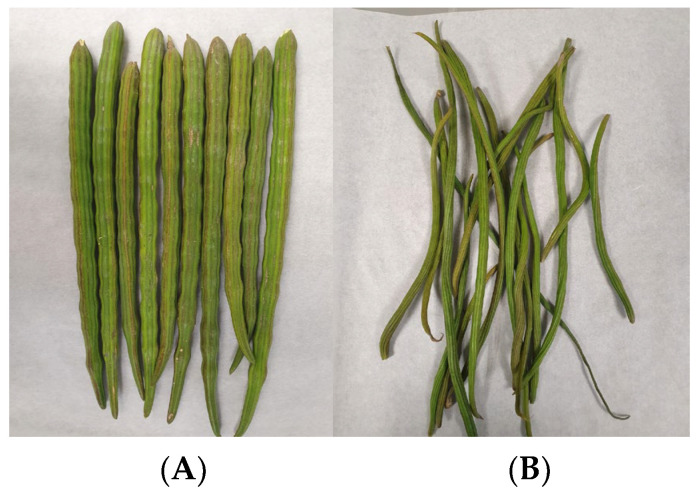
Mature (**A**) and tender (**B**) moringa pods.

**Figure 2 foods-13-01823-f002:**
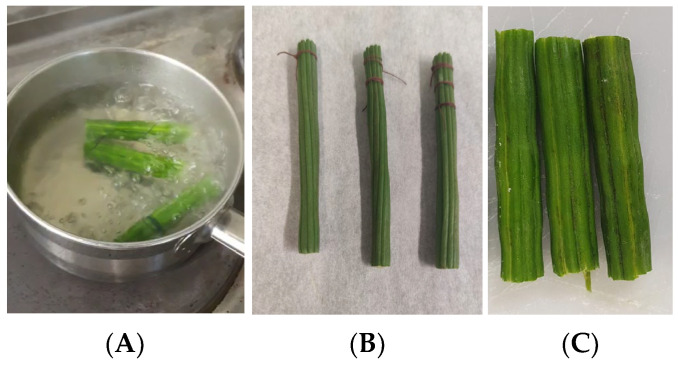
Boiling procedure (**A**); appearance of tender (**B**) and mature (**C**) moringa pod after 20 min of boiling.

**Figure 3 foods-13-01823-f003:**
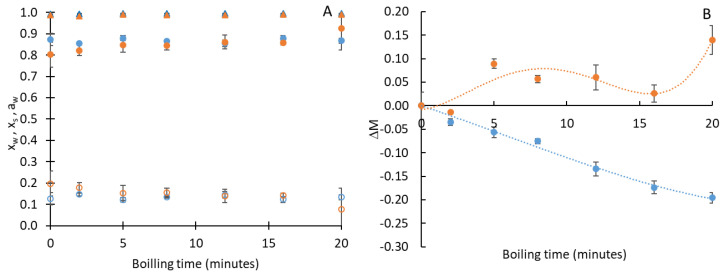
Mass fraction of water (x_w_: g water/g pod) (filled circles), solutes (x_s_: g solutes/g pod) (empty circles) and water activity (a_w_) (triangles) (**A**) and total mass variation (**B**) in *Moringa oleifera* pods versus cooking times and in different ripening states: mature (blue symbols) and tender (orange symbols).

**Figure 4 foods-13-01823-f004:**
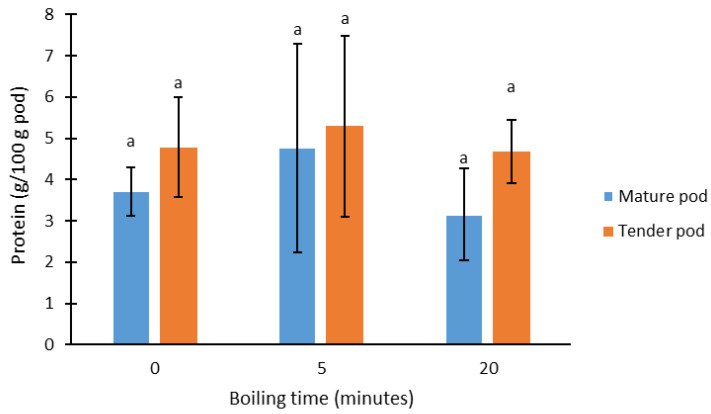
Protein content in moringa pods according to cooking time and maturity stage. Same letters indicate homogeneous groups obtained in the ANOVA with a significance level of 95%.

**Figure 5 foods-13-01823-f005:**
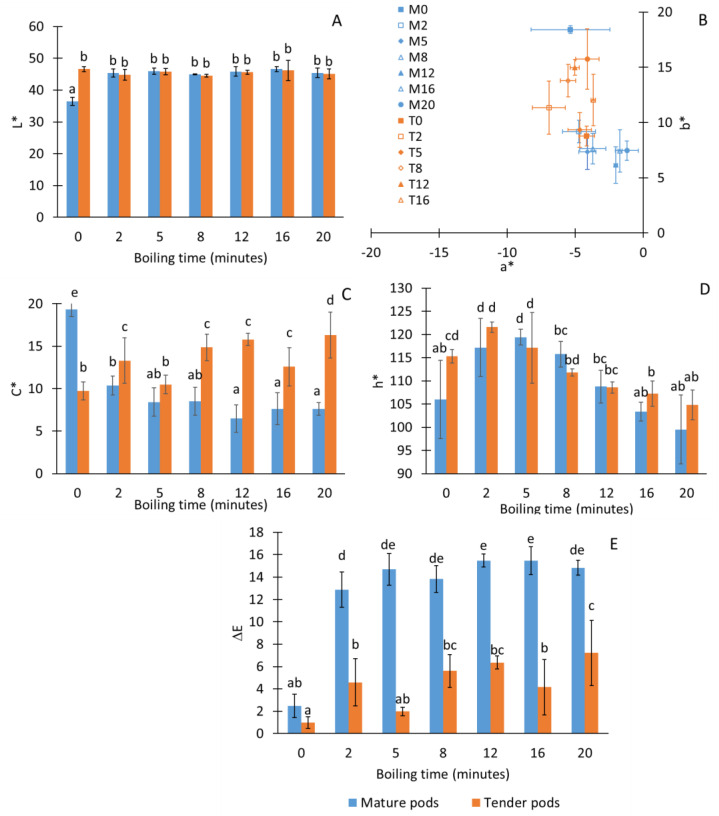
Luminosity (L*) (**A**), location in the chromatic plane of the a* and b* coordinates (**B**), chrome (**C**), hue (**D**) and difference of color with regards to raw material (**E**) of the moringa pods according to the cooking time and maturity stage. Numbers in the legend indicate the boiling times. Same letters indicate homogeneous groups obtained in the ANOVA with a significance level of 95%.

**Figure 6 foods-13-01823-f006:**
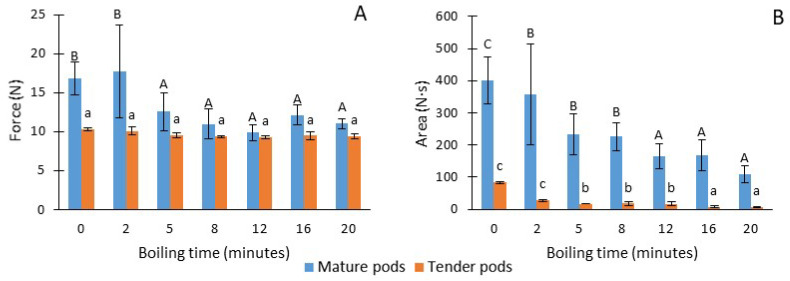
Maximum force (**A**) and area (**B**) of the puncture test on pods according to cooking time and maturity stage. Same letters indicate homogeneous groups obtained in the ANOVA with a significance level of 95%.

**Figure 7 foods-13-01823-f007:**
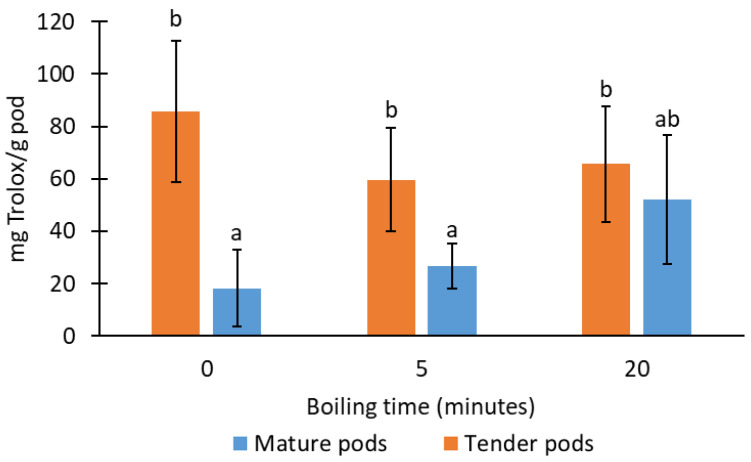
Antioxidant capacity of moringa pods according to the cooking time and maturity stage. Same letters indicate homogeneous groups obtained in the ANOVA with a significance level of 95%.

**Figure 8 foods-13-01823-f008:**
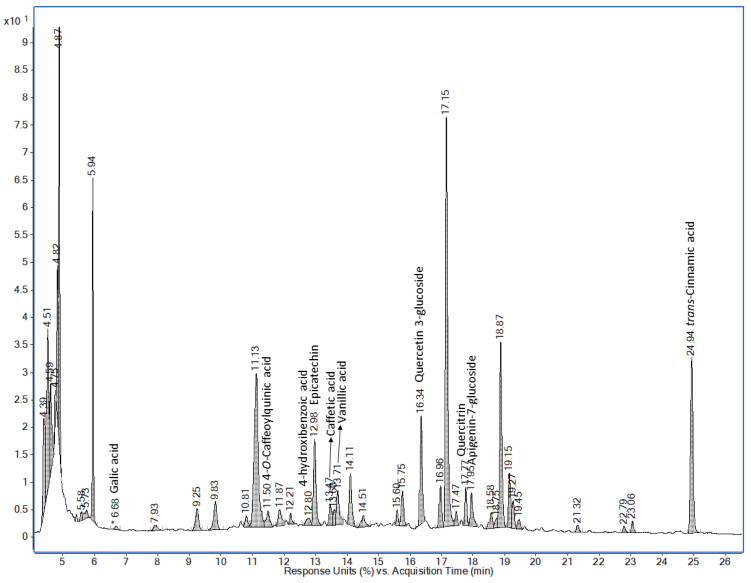
Example of HPLC chromatograph of the phenolic profile in tender moringa pods boiled for 5 min.

**Figure 9 foods-13-01823-f009:**
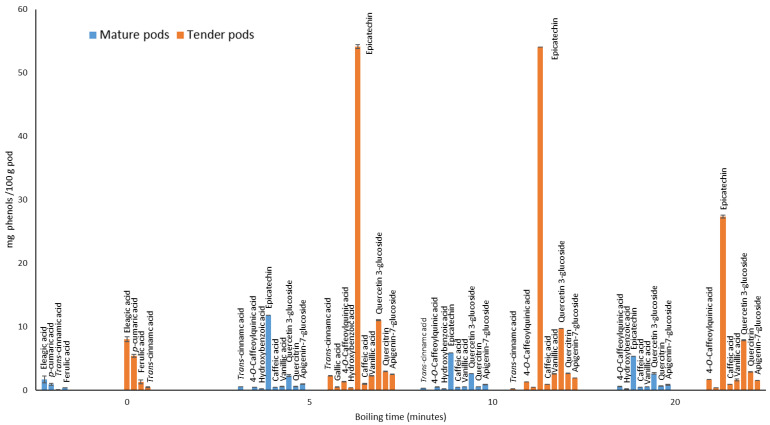
Phenols present in the pods and their evolution after different cooking times, taking into account the maturity stage.

**Table 1 foods-13-01823-t001:** Gradient elution program used in the study.

Time (Minutes)	% A
0	90
3	85
18	60
24	60
27	34
27	50
33	30
40	10
43	90
45	90

**Table 2 foods-13-01823-t002:** Phenolic content (average values and standard deviation) in tender and mature *Moringa oleifera* pods according to different cooking times (0, 5, 10 and 20 min) at 100 °C. The data are expressed in mg/100 g of pod (*n* = 3).

			Cooking Time					
	0 min		5 min		10 min		20 min	
	Mature Pods	Tender Pods	Mature Pods	Tender Pods	Mature Pods	Tender Pods	Mature Pods	Tender Pods
*phenolic acids*								
Ellagic acid	1.715 (0.56)	8.047 (0.39)	n.d.	n.d.	n.d.	n.d.	n.d.	n.d.
*p*-Coumaric acid	0.956 (0.95)	5.396 (0.25)	n.d.	n.d.	n.d.	n.d.	n.d.	n.d.
Ferulic acid	0.429 (0.43)	1.338 (0.28)	n.d.	n.d.	n.d.	n.d.	n.d.	n.d.
*Trans*-cinnamic acid	0.111 (0.02)	0.480 (0.08)	0.566 (0.0009)	2.279 (0.03)	0.299 (0.013)	0.249 (0.0005)	n.d.	n.d.
Gallic acid	n.d.	n.d.	n.d.	0.527 (0.02)	n.d.	n.d.	n.d.	n.d.
4-*O*-caffeoylquinic acid	n.d.	n.d.	0.475 (0.004)	1.375 (0.02)	0.535 (0.0106)	1.328 (0.014)	0.611 (0.01)	1.710 (0.015)
4-Hydroxybenzoic acid	n.d.	n.d.	0.236 (0.0005)	0.439 (0.010)	0.226 (0.0005)	0.469 (0.003)	0.244 (0.03)	0.432 (0.0003)
Caffeic acid	n.d.	n.d.	0.454 (0.002)	1.048 (0.06)	0.456 (0.0002)	0.934 (0.003)	0.461 (0.001)	0.931 (0.0012)
Vanillic acid	n.d.	n.d.	0.624 (0.04)	2.316 (0.04)	0.569 (0.02)	2.625 (0.04)	0.576 (0.03)	1.664 (0.17)
*flavonoids*								
Quercetin 3-glucoside	n.d.	n.d.	2.348 (0.17)	11.033 (0.09)	2.616 (0.006)	9.725 (0.03)	2.68 (0.15)	7.868 (0.08)
Quercitrin	n.d.	n.d.	0.610 (0.013)	2.978 (0.04)	0.586 (0.0012)	2.688 (0.02)	0.673 (0.03)	2.922 (0.03)
Apigenin-7-glucoside	n.d.	n.d.	0.968 (0.020)	2.526 (0.019)	0.913 (0.005)	1.967 (0.011)	0.896 (0.07)	1.595 (0.006)
Epicatechin	n.d.	n.d.	11.797 (0.012)	54.135 (0.28)	5.897 (0.006)	54.075 (0.0003)	5.334 (0.02)	27.346 (0.2)

n.d.: not detected.

## Data Availability

The original contributions presented in the study are included in the article, further inquiries can be directed to the corresponding author.

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
