# Peer review of "Influence of Boiling Time on Chemical Composition and Properties of Tender and Mature Moringa Pods"

_foods, 2024, doi:10.3390/foods13121823_

Round 1
Reviewer 1 Report
Comments and Suggestions for Authors
The study titled "Influence of boiling time on properties of tender and mature moringa pods" discussed the notable influence of cooking time on two distinct stages of moringa pods, producing new polyphenols during the cooking process. While this paper provides valuable insights on the optimal cooking time, there is room for further enhancement.
Abstract: Specify the boiling temperature and storage duration
Introduction: Provide background information on the cooking, boiling, or steaming of Moringa Pods
Materials and Methods:
Line 86: Include details on the harvesting of tender and mature pods, including the date and month
Results: Consider labeling the treatment on the graph with a caption, such as a blue color bar indicating the mature pod for easier readability by reviewers. Ensure that the p in p-Coumaric acid, trans in cinnamic acid, and o in 4-O-caffeoylquinic acid are italicized.
The discussion section requires improvement and should be separated from the results for clarity.
Comments on the Quality of English Language
Need grammar check carefully
Reviewer 2 Report
Comments and Suggestions for Authors
Dear authors, my remarks and comments are in the attached file.

Reviewer 3 Report
Comments and Suggestions for Authors
Suggest changes the title “Influence of boiling time on properties of tender and mature moringa pods” to “Effect of cooking process on the chemical composition and properties of tender and mature moringa”
In abstracts:
- The goal is unclear. It should match the one you placed at the end of the introduction.
Introduction
Quite paragraph de “Current agri-food …to … them unsustainable”
Add preservation method information.
Talk more about the "maturation process"
Methodology:
In section 2.2. What did you base your cooking times on, and what is your control?
Why were they cut into strips? In the end, if it's a method of preservation, didn't they consider boiling the whole pod?
Place pictures of how the pod looks after cooking.
Place references in the different methodologies
In section 2.3., Why was the Aw (water activity) not determined, is a value that would support the effect of cooking.
It is recommended to measure the size of the pods, because of the cooking process.
In HPLC, why was this binary mobile phase used?
Line 158 to 161: Why wasn't some microscopy technique added that would allow you to identify what the changes are generated by the cooking process?
It is recommended to change "boilling time" to "cooking time."
Throughout the document, replace "stage of maturity" with "maturity stage"
Check English
Reviewing and delving into the discussion of color change, I think it goes more for the changes in its composition, since the firing process can cause oxidation of components or pigments.
To support with more results his statement of lines 208 to 212.
Why is the antioxidant capacity only displayed at 0, 5 and 20 min cooking time?
In the figures, indicate what it means to have the same letters, e.g., that there are no significant differences, etc.
According to their Figure 5, there are no significant changes in the antioxidant capacity of tender pods, compared to mature pods. Explain in more detail.
In the discussion of Table 2: how much influence the state of ripeness, I doubt very much that the cooking time has favored the synthesis, perhaps it is important to discuss it in the document. It could have been that the cooking process generated microstructural changes, which led to greater availability, or in their case some could have oxidized, etc. Dig deeper into this section.
Was the cooking water evaluated? This could lead to phenolic compounds that were solubilized in the water.
What is the sensitivity of HPLC?
It is recommended to make a diagram.
In general:
- Check English
- There is no References section
- Replace or standardize the term "cooking process" in the document
- Their discussions need to be expanded.
- It remains to be mentioned how after the process of cooking, how it would be stored to be able to keep longer.
Comments on the Quality of English Language
Improve your English
Round 2
Reviewer 1 Report
Comments and Suggestions for Authors
Accepted
Comments on the Quality of English Language
checking Typo and grammer